# High BMI and Surgical Time Are Significant Predictors of Lymphocele after Robot-Assisted Radical Prostatectomy

**DOI:** 10.3390/cancers15092611

**Published:** 2023-05-04

**Authors:** Simon Gloger, Christian Wagner, Sami-Ramzi Leyh-Bannurah, Stefan Siemer, Madeleine Arndt, Jens-Uwe Stolzenburg, Toni Franz, Burkhard Ubrig

**Affiliations:** 1Center for Minimally Invasive and Robotic Urology, Augusta Hospital Bochum, Witten/Herdecke University, Bergstraße 26, 44791 Bochum, Germany; 2Department of Urology, Pediatric Urology and Urologic Oncology, St. Antonius Hospital Gronau, Möllenweg 22, 48599 Gronau, Germany; 3Department of Urology, Saarland University, Kirrberger Str. 100, 66421 Homburg, Germany; 4Department of Urology, University of Leipzig, Liebigstraße 20, 04103 Leipzig, Germany

**Keywords:** robot-assisted radical prostatectomy, lymphocele, symptomatic lymphocele, risk factors, pelvic lymph node dissection, RARP, peritoneal flap

## Abstract

**Simple Summary:**

Robot-assisted prostatectomy is one of the main therapeutic options for localized prostate cancer. Lymphoceles are one of the major causes of complications after robot-assisted radical prostatectomy and pelvic lymph node dissection. Because the data on risk factors for lymphoceles remains unclear and inconclusive, they were investigated in the present study using the data set of the prospective randomized trial ProLy. In this study, high BMI was found to be an independent predictor for the occurrence of lymphoceles and symptomatic lymphoceles. In addition, a longer surgical time proved to be another risk factor for the postoperative occurrence of lymphoceles.

**Abstract:**

Lymphoceles (LC) occur in up to 60% after robot-assisted radical prostatectomy (RARP) and pelvic lymphadenectomy (PLND). In 2–10%, they are symptomatic and may cause complications and require treatment. Data on risk factors for the formation of lymphoceles after RARP and PNLD remain sparse in the urologic literature and are inconclusive to date. The underlying data of this secondary analysis were obtained from the prospective multi-center RCT ProLy. We performed a multivariate analysis to focus on the potential risk factors that may influence lymphocele formation. Patients with LC had a statistically significant higher BMI (27.8 vs. 26.3 kg/m^2^, *p* < 0.001; BMI ≥ 30 kg/m^2^: 31 vs. 17%, *p* = 0.002) and their surgical time was longer (180 vs. 160 min, *p* = 0.001) In multivariate analysis, the study group (control vs. peritoneal flap, *p* = 0.003), BMI (metric, *p* = 0.028), and surgical time (continuous, *p* = 0.007) were independent predictors. Patients with symptomatic lymphocele presented with higher BMI (29 vs. 26.6 kg/m^2^, *p* = 0.007; BMI ≥ 30 kg/m^2^: 39 vs. 20%, *p* = 0.023) and experienced higher intraoperative blood loss (200 vs. 150 mL, *p* = 0.032). In multivariate analysis, BMI ≥ 30 kg/m^2^ vs. < 30 kg/m^2^ was an independent predictor for the formation of a symptomatic lymphocele (*p* = 0.02). High BMI and prolonged surgical time are general risk factors for the development of LC. Patients with a BMI ≥ 30 kg/m^2^ had a higher risk for symptomatic lymphoceles.

## 1. Introduction

Bilateral pelvic lymphadenectomy (PLND) is currently recommended as an adjunct to radical robot-assisted prostatovesiculectomy (RARP) for medium- and high-risk prostate carcinomas [1,2]. One of the principal complications of lymphadenectomy is lymphoceles which has been reported in 24–51% of patients after RARP with PLND [3,4,5].

Whilst most lymphoceles (LC) remain asymptomatic and need no treatment, symptomatic lymphoceles (sLC) require treatment and occur in 2–10% of patients [6]. Typical complications from sLCs include superinfection, leg swelling, deep vein thrombosis, and hydronephrosis [7]. Symptomatic lymphoceles are among the most common reasons for readmission after RARP and PLND [8]. Moreover, most studies have focused on the open retropubic-extraperitoneal surgical approach [9,10,11,12,13].

In recent years, numerous authors have proposed surgical strategies for reducing the incidence of lymphoceles [14,15,16,17,18]. Several groups have examined the effect of bilateral peritoneal flaps (PIFs), which are suture-fixated to the endopelvic fascia, on the formation of LC and sLC [5,6,19,20,21,22].

A prospective randomized multicenter double-blind ProLy trial has recently compared patients against those without a PIF and found that PIFs resulted in a significant reduction in the overall incidence of LCs (33% to 22%) and in the incidence of sLCs (8.1% to 3.3%) [7].

The present study aimed to identify potential clinical risk factors for LCs and sLCs through a secondary analysis of the ProLy trial data.

## 2. Methods

### 2.1. Study Group

We used the data set from the prospective randomized, double-blind ProLy study [7]. After dropout and loss-to-follow-up, the ProLy study analyzed a total of 475 patients. Data were available for the statistical analysis of this population during the recruitment period from November 2018 to August 2020. The vote of the ethics committee (Aerztekammer Muenster, AZ 2018-451-f-S), as well as the registration in the German Register of Clinical Trials (registration number DRKS00015720), was obtained before the start of the study and covered secondary data analysis.

### 2.2. Surgical Technique

All RARPs were performed transperitoneal. PLND and the formation of bilateral PIFs were standardized in the study protocols. Randomization was performed intraoperatively after the completion of RARP and PLND. According to the randomization result, the patients had either received PIF or not.

### 2.3. Data Acquisition and Follow-Up

#### 2.3.1. Preoperative Data Set (Patient-Related Risk Factors)

Patient-related parameters had been prospectively collected at study enrollment and before randomization. The following preoperative data were available for secondary analysis: Body mass index (BMI), prostate volume at admission, initial PSA, American Society of Anesthesiologists (ASA) score, and the Charlson Comorbidity Index (CCI).

#### 2.3.2. Perioperative Data Set (Procedure- and Surgeon-Related Risk Factors)

These included operative time, intraoperative blood loss, intraoperative drain placement, length of hospital stay, the total number of harvested lymph nodes, Gleason grade, ISUP (International Society of Urological Pathology) group, pT stage, pN and R status.

#### 2.3.3. Postoperative Follow-Up

Two follow-up appointments were performed postoperatively at 30 and 90 days. These comprised a thorough history focusing on postoperative complications and abdominal and pelvic ultrasonography with a focus on pelvic lymphocele. Follow-up was performed not by the surgical team but by other physicians trained in ultrasound examinations. The study protocol included daily subcutaneous injections of low-molecular-weight heparin for 4 weeks postoperatively.

### 2.4. Statistical Analysis

The ProLy data set was analyzed for risk factors contributing to the formation of LC and sLCs. Corresponding analyses were performed for the entire sampled population, the peritoneal flap group, and the control group; to describe the metric variables, we determined the count, median, and interquartile range. The Shapiro–Wilk test was used for these variables to check if they were normally distributed. If the normal distribution assumption was not rejected (*p*-value < 0.1), a comparison was performed using the t-test. If it was rejected, we applied the Mann–Whitney U test. The distribution of categorical data was described in terms of absolute and relative frequencies. We compared the frequency distributions of a categorical variable of independent groups using the χ^2^ square test or Fisher‘s exact test in case of expected cell frequencies of less than five.

Next, we performed univariate and multivariate analyses to determine potential predictors for the formation of lymphocele and of symptomatic lymphocele. A possible center effect with respect to lymphoceles or symptomatic lymphoceles could be excluded before conducting further analysis. The influence of BMI as one possible predictor was calculated in two different models: both continuously and dichotomized according to obesity as defined (≥30 kg/m^2^ vs. <30 kg/m^2^). Due to only minimal deviations from the study protocol, no analysis with respect to different heparin use was performed. Multivariate analysis was performed as a stepwise regressive elimination of the independent variables. Because of the small number of events in only eight symptomatic lymphoceles in the peritoneal flap group, we skipped performing logistic regression for this subgroup. All tests were conducted two-sided, and the significance level was set at *p* < 0.05. For all data analyses, Dr. Silke Lange (University of Witten/Herdecke, Germany), a certified biostatistician, SAS^®^ 9.4 statistical software, was used (SAS Institute, Cary, NC, USA).

## 3. Results

All 475 records from the ProLy study were eligible for analysis. A total of 239 patients had received the peritoneal flap, while 236 patients formed a control group.

### 3.1. Incidence of Lymphoceles (LCs)

Table 1 shows the frequency distribution of the above-described pre- and perioperative potential risk factors, broken down by patients with and without LC formation: patients with LC presented a statistically significantly higher BMI (27.8 vs. 26.3 kg/m^2^, *p* < 0.001). They were more frequently clinically obese with a BMI of more than 30 kg/m^2^ (31 vs. 17%, *p* = 0.002). Patients with LC had a significantly longer surgical time of approximately 20 min (*p* = 0.001) and a longer hospital stay (0.3 days; *p* = 0.02).

We also separately analyzed the treatment group (with PIF) and the control group (without PIF). In the PIF group, BMI (27.6 vs. 26.4 kg/m^2^, *p* = 0.03), obesity (31% vs. 16%, *p* = 0.02) and surgical time (182 vs. 152 min; *p* = 0.02) were statistically significant, whereas the length of hospital stay was not (7.6 days vs. 7 days, *p* = 0.15). In the control group, only a higher BMI (28.8 vs. 26.6 kg/m^2^, *p* = 0.03) was statistically and significantly associated with lymphocele formation.

### 3.2. Multivariate Analysis (LCs)

In the multivariate analysis of the total study sample (Table 2), randomization (PIF vs. control; *p* = 0.003), BMI (metric; *p* = 0.03), and surgical time (*p* = 0.007) turned out to be independent predictors for LC formation. When separately analyzing the PIF group, only longer surgical time was a predictor of LCs (*p* = 0.02); in the control group, BMI was an independent predictor of LC formation (metric; *p* = 0.008).

### 3.3. Incidence of Symptomatic Lymphoceles (sLCs)

sLCs occurred in only 3.3% of the PIF and in 8.1% of the control group. Table 3 shows the frequency distribution of clinical parameters correlated with symptomatic lymphoceles (sLCs). In the total study sample, patients with symptomatic lymphocele had statistically and significantly higher BMI values (29 vs. 26.6 kg/m^2^, *p* = 0.007) and a higher proportion of obesity (BMI > 30 kg/m^2^, 39 vs. 20, *p* = 0.02). Additionally, the documented intraoperative blood loss was found to be higher (200 vs. 150 mL, *p* = 0.03) compared to patients without symptomatic lymphocele. In the subgroup analysis, with respect to sLCs, none of the clinical parameters reached significant levels in the PIF group. In the control group, only the BMI of patients with a symptomatic lymphocele was significantly higher than that of patients without a symptomatic lymphocele (28.8 vs. 26.6 kg/m^2^, *p* = 0.03).

### 3.4. Multivariate Analysis (sLC)

Obesity (BMI ≥ 30 kg/m^2^) was an independent predictor for the formation of a symptomatic lymphocele in the total study sample (*p* = 0.02) (Table 4). In the control group, this was applied to blood loss (*p* = 0.04). Due to the low incidence (*n* = 8, 3.3%) of sLC in the PIF group, we did not perform a multivariate analysis on this subgroup.

## 4. Discussion

The present study used data from the prospective randomized ProLy study to investigate risk factors for lymphocele formation after RARP and PLND. In the ProLy study collective, the overall incidence of the lymphocele (LC) was 28% among 475 patients during a follow-up period of 90 days, with 5.7% of the patients developing a symptomatic lymphocele (sLC) [7].

The identification of risk factors seems important for identifying and counseling patients prior to surgery. To date, data on risk factors for lymphocele after RARP and PLND remain scarce and inconclusive in the urologic literature. Furthermore, most of this work relates to the open surgical retropubic approach [9,12,13,23,24].

For robot-assisted RARP, Goßler et al. were able to show an increased risk of lymphocele formation in the presence of a high BMI (OR 1.15, CI 1.05–1.27) in obese patients (OR 2.76, CI 1.05–7.3), aggressive tumors (OR 3.25, CI 1.18–8.92) and an extended operative time (OR 1.01, CI 1.01–1.02) [25]. Another study by Sforza et al. presented higher occurrences of sLC in men with higher BMI (30.4 vs. 25.8 kg/m^2^) and a history of vascular surgery (23.8% vs. 8.4%), but only a higher BMI was an independent predictor of sLC after performing multivariate analysis [26]. Similar to the ProLy study, the recently published PerFix study revealed lower incidences of sLC for patients receiving a peritoneal flap (11.5 vs. 2.4%). In their study BMI (OR 1.1, CI 1.03–1.26) and intervention (OR 4.6, CI 1.28–16.82) were independent risk factors for sLC on multivariate analysis [20]. Magistro et al. combined retrospective cohorts of the robotic and retropubic approach of radical prostatectomy and described an increased risk of patients with poorly differentiated tumors (Gleason Score ≥ 8) or elevated PSA levels (≥10 ng/mL) as well as for patients with extensive lymphadenectomy (lymph node yield ≥ 11) [10].

In our patient sample, we found a significantly increased risk for the formation of lymphoceles in patients with an elevated BMI. This risk factor was valid for the overall incidence of lymphoceles (OR 1.07, CI 1.01–1.13, Table 2) as well as for the occurrence of sLC. Obese patients—when using a BMI ≥30 kg/m^2^—had almost triple the risk (OR = 2.86, CI 1.18–6.9; *p* = 0.02) of forming a symptomatic lymphocele (Table 4). This result was consistent with the abovementioned, prospective RCT by Goßler et al. who also found an almost three times higher risk for obese patients (BMI ≥ 30 kg/m^2^) in developing sLC [25]. Student et al. and Sforza et al. confirmed this finding with their multivariate analyses, demonstrating an OR of 1.1 and 1.7, respectively [20,26]. Intraoperative challenging conditions and chronic inflammation in the adipose tissue could provide reasons for the higher risk of postoperative LC and sLC [25,26,27,28]. BMI was also found to be an independent predictor of Clavien 3 complications in patients undergoing RARP with extended PLND [29]. Symptomatic lymphoceles as a major cause of high-grade Clavien-Dindo complications may have been a contributing factor. On the other hand, a study by Mundhenk et al. contrasted with these data, showing that patients with a low BMI had an increased risk of developing symptomatic lymphocele [30]. However, besides being a retrospective report, a limitation of their study is that it did not differentiate between open surgical and transperitoneal robotic-assisted access.

In our study, in addition to BMI, we found increased surgical time to be an independent predictor of lymphocele formation (Table 2). On average, this was 20 min longer (180 vs. 160 min) in patients with the formation of LC than those without. This may indicate a more challenging dissection as potentially favoring lymphocele formation. Our findings are also supported by the already aforementioned study by Goßler and coworkers, who documented this risk factor for symptomatic lymphoceles. They observed an average prolongation of surgical time by 15 min (165 vs. 180 min) in their study sample [25]. Intraoperative blood loss could also be indicative of a more difficult dissection. In our study population, we found significantly higher blood loss in patients who had formed symptomatic lymphocele (Table 3).

The aim of the ProLy study was to investigate the effect of a surgically constructed peritoneal flap (PIF) on lymphocele incidence. The study found that patients with PIF formed significantly fewer LCs (22 vs. 33%) and sLCs (3.3 vs. 8.1%) [7]. As to be expected, on multivariate analysis, the study group was found to be an independent risk factor for the occurrence of LC. Compared to the peritoneal flap group, the control group had a nearly doubled risk (OR 1.99, CI 1.26–3.15; *p* = 0.003) of forming a lymphocele (Table 2). However, on risk factor analysis for the development of sLC, the assignment to the different study group revealed an increased risk for the control group on univariate analysis but did not reach a level of significance on multivariate analysis (Table 4). These data were supported by the PerFix study: Student et al. found the study group to be an independent predictor of symptomatic lymphocele formation with an odds ratio of 4.6 [20]. The study by Goßler et al. differed on this point. Here, the peritoneal flap was not an independent predictor for the development of a symptomatic lymphocele [25].

Additional risk factors such as patient age, the length of lymphadenectomy/number of lymph nodes removed, Gleason grade, or positive nodal status were reported inconsistently and have been discussed controversially in the literature [3,4,11,23,29,30,31]. In our study, we could not confirm these potential risk factors.

Patients with an ASA score of 2/3 were shown to have a higher incidence of Clavien–Dindo complications greater than one [32]. We also looked at this in our study with respect to lymphoceles or symptomatic lymphoceles as a major cause for high-grade Clavien–Dindo complications but could not confirm this either.

## 5. Limitations

The main limitation of this data analysis was possibly the small number of symptomatic lymphoceles, which occurred in only 27 patients (5.7%) in the underlying ProLy study. Independent predictors of symptomatic lymphocele may not have been identified due to this small number. Moreover, owing to a small number of symptomatic lymphoceles, multivariate subgroup analysis could not be performed on the peritoneal flap group. Large-scale studies with a higher number of patients would certainly be useful in this regard. Furthermore, the median lymph node yield of the ProLy study was relatively low (14 in each group). Comparable studies had slightly higher lymph node yields (14–17 lymph nodes) [22,33,34,35]. This may have affected the results and conclusions of the study.

## 6. Conclusions

In this secondary analysis of the ProLy data set, we found two important independent predictors for lymphocele formation and one for symptomatic lymphocele formation. Obese patients with a BMI ≥ 30 kg/m^2^ had a higher risk of symptomatic lymphoceles. Increasing BMI was also correlated with a higher risk for the development of LC. In addition, increasing time spent in surgery was associated with higher incidences of lymphoceles. This could be explained by more challenging surgery conditions resulting in more adverse postoperative events. Therefore, in these patient groups, a close follow-up should be performed in the subsequent postoperative stay and ambulatory care to detect possible complications at an early stage.

## Figures and Tables

**Table 1 cancers-15-02611-t001:** Frequency distribution of the potential risk factors for the occurrence of a lymphocele.

Potential Risk Factor	Patients w/o Lymphocele (*n* = 338)	Patients with Lymphocele (*n* = 129)	*p*
Preoperative variables			
Age (Years, IQR)	66 (60–70)	65 (60–70)	0.8 ^a^
BMI (kg/m^2^, IQR)	26.3 (24.4–28.7)	27.8 (25.8–30.4)	<0.001 ^a^
BMI ≥ 30 kg/m^2^ (*n*, %)	58 (17)	39 (30)	0.002 ^b^
Prostate volume (mL, IQR)	40 (30–50)	41 (30–60)	0.08 ^a^
PSA (ng/mL, IQR)	7.2 (5.2–11)	7.2 (5.3–10.3)	0.8 ^a^
Median CCI (SD)	2.2 (0.5)	2.3 (0.5)	0.4 ^a^
ASA Score >1 (*n*, %)	284 (86)	113 (88)	0.6 ^b^
ASA Score 3 (*n*, %)	42 (13)	18 (14)	0.7 ^b^
Intra- and perioperative variables			
Surgery time (minutes, IQR)	160 (140–182)	180 (150–200)	0.001 ^a^
Blood loss (mL, IQR)	150 (100–250)	150 (100–250)	0.2 ^a^
Inpatient length of stay (days, SD)	7.2 (5.3)	7.5 (4.4)	0.02 ^a^
Drainage (*n*, %)	65 (20)	21 (16)	0.5 ^b^
Pathological variables			
Total lymph nodes (*n*, %)	14 (11–19)	14 (11–18)	0.5 ^b^
ISUP-GGG ≥3 (*n*, %)	132 (40)	60 (47)	0.15 ^b^
T-stage >pT2 (*n*, %)	141 (43)	47 (38)	0.3 ^b^
Nodal status pN1 (*n*, %)	25 (8)	10 (8)	0.9 ^b^
R1status (*n*, %)	35 (11)	11 (9)	0.5 ^b^

ASA = American Society of Anesthesiologists, BMI = Body mass index, CCI = Charlson comorbidity index, ISUP-GGG = International Society of Urological Pathology—Gleason grading groups, IQR = interquartile range, PSA = prostate-specific antigen, SD = standard deviation. ^a^ Mann–Whitney U test; ^b^ χ^2^ square test.

**Table 2 cancers-15-02611-t002:** Univariate and multivariate analysis of potential risk factors for the occurrence of lymphocele.

	Univariate				Multivariate			
		95% CI				95% CI		
Potential Risk Factor	OR	Lower Bound	Upper Bound	*p*	OR	Lower Bound	Upper Bound	*p*
Preoperative variables								
Age (Years, continuous)	0.99	0.97	1.03	0.9				
BMI (metric)	1.1	1.04	1.15	<0.001	1.07	1.01	1.13	0.03
BMI (≥30kg/m^2^ vs. <30kg/m^2^)	2.12	1.32	3.39	0.002				
Prostate volume (mL, continuous)	1.01	1	1.02	0.049				
PSA (ng/mL, continuous)	0.99	0.98	1.01	0.7				
CCI (≥1 vs. 0)	1.12	0.78	1.63	0.5				
ASA Score (>1 vs. 1)	1.17	0.64	2.15	0.6				
ASA Score (3 vs. <3)	1.12	0.62	2.02	0.7				
Intra- and perioperative variables								
Surgery time (minutes, continuous)	1.01	1.004	1.01	<0.001	1.01	1.002	1.01	0.007
Blood loss (mL, continuous)	1	1	1.002	0.1				
Length of stay (days, continuous)	1.01	0.97	1.05	0.5				
Study group (Control vs. peritoneal flap)	1.75	1.16	2.64	0.008	1.99	1.26	3.15	0.003
Drainage (yes vs. no)	0.82	0.48	1.4	0.5				
Pathological variables								
Total lymph nodes (continuous)	0.99	0.96	1.03	0.6				
ISUP-GGG (3–5 vs. 1–2)	1.02	0.85	1.23	0.8				
T-stage (>pT2 vs. pT2)	0.82	0.54	1.25	0.4				
Nodal status (pN1 vs. pN0)	1.06	0.49	2.27	0.9				
R-status (R1 vs. R0)	0.9	0.52	1.58	0.7				

**Table 3 cancers-15-02611-t003:** Description of patients with and without symptomatic lymphocele.

Potential Risk Factor	Pat. w/o Sympt. Lymphocele (*n* = 448)	Pat. with Sympt. Lymphocele (*n* = 27)	*p*
Preoperative variables			
Age (years, IQR)	66 (60–70)	66 (63–72)	0.3 ^a^
BMI (kg/m^2^, IQR)	26.6 (24.5–29.2)	29 (26.2–32)	0.007 ^a^
BMI ≥ 30 kg/m^2^ (*n*, %)	88 (20)	10 (39)	0.023 ^b^
Prostate volume (mL, IQR)	40 (30–53)	41 (30–55)	0.7 ^a^
PSA (ng/mL, IQR)	7.2 (5.3–10.6)	7.3 (4.8–14.4)	0.9 ^a^
Median CCI (SD)	2.3 (0.5)	2.3 (0.5)	0.3 ^a^
ASA Score >1 (*n*, %)	381 (86)	22 (85)	0.8 ^c^
ASA Score 3 (*n*, %)	59 (13)	2 (7.7)	0.4 ^c^
Intra- and perioperative variables			
Surgery time (min, IQR)	160 (140–190)	151 (150–182)	0.6 ^a^
Blood loss (mL, IQR)	150 (100–250)	200 (150–300)	0.032 ^a^
Length of stay (days, SD)	7.3 (5.2)	7 (2.3)	0.3 ^a^
Drainage (*n*, %)	81 (18)	5 (19)	>0.9 ^c^
Pathological variables			
Total lymph nodes (*n*, %)	14 (11–18)	13 (11–20)	0.8 ^b^
ISUP-GGG ≥ 3 (*n*, %)	181 (41)	13 (50)	0.4 ^b^
T-stage >pT2 (*n*, %)	181 (41)	10 (44)	0.8 ^b^
Nodal status pN1 (*n*, %)	33 (10)	2 (8)	>0.9 ^c^
R1-Status (*n*, %)	46 (14)	2 (8)	>0.9 ^c^

ASA = American Society of Anesthesiologists, BMI = Body mass index, CCI = Charlson comorbidity index, ISUP-GGG = International Society of Urological Pathology—Gleason grading groups, IQR = interquartile range, PSA = prostate-specific antigen, SD = standard deviation. ^a^ Mann–Whitney U test; ^b^ χ^2^ square test; ^c^ Fisher’s exact test.

**Table 4 cancers-15-02611-t004:** Univariate and multivariate analysis of potential risk factors for the occurrence of symptomatic lymphocele.

	Univariate				Multivariate			
		95% CI				95% CI		
Potential Risk Factor	OR	Lower Bound	Upper Bound	*p*	OR	Lower Bound	Upper Bound	*p*
Preoperative variables								
Age (years, continuous)	1.02	0.97	1.08	0.4				
BMI (metric)	1.08	0.99	1.18	0.054				
BMI (≥30 kg/m^2^ vs. <30 kg/m^2^)	2.53	1.11	5.76	0.027	2.86	1.18	6.9	0.02
Prostate volume (mL, continuous)	1.002	0.98	1.02	0.9				
PSA (ng/mL, continuous)	1.01	0.99	1.03	0.3				
CCI (≥1 vs. 0)	1.19	0.61	2.35	0.6				
ASA Score (>1 vs. 1)	0.87	0.29	2.6	0.8				
ASA Score (3 vs. <3)	0.54	0.12	2.34	0.4				
Intra- and perioperative variables								
Surgery time (min, continuous)	1	0.99	1.01	0.7				
Blood loss (mL, continuous)	1	0.99	1.002	0.3				
Length of stay (days, continuous)	0.98	0.88	1.09	0.7				
Study group (control vs. peritoneal flap)	2.53	1.08	5.9	0.032				
Drainage (yes vs. no)	1.03	0.38	2.8	0.9				
Pathological variables								
Total lymph nodes (continuous)	1.01	0.94	1.08	0.9				
ISUP-GGG (3–5 vs. 1–2)	0.98	0.68	1.4	0.9				
T-stage (>pT2 vs. pT2)	1.11	0.48	2.59	0.8				
Nodal status (pN1 vs. pN0)	1.03	0.23	4.53	0.9				
R-Status (R1 vs. R0)	0.65	0.17	2.55	0.5				

ASA = American Society of Anesthesiologists, BMI = Body mass index, CCI = Charlson comorbidity index, ISUP-GGG = International Society of Urological Pathology—Gleason grading groups, CI = confidence interval, OR = odds ratio, PSA = prostate-specific antigen.

## Data Availability

The data presented in this study are available on request from the corresponding author.

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
