# Peer review of "High BMI and Surgical Time Are Significant Predictors of Lymphocele after Robot-Assisted Radical Prostatectomy"

_cancers, 2023, doi:10.3390/cancers15092611_

Round 1

Reviewer 1 Report

This paper reports the data analysis results of the multi-center randomized trial ProLy. I have the following questions about and comments on this article.

Major:

1. In the trial, since the randomization is conducted after the completion of RARP and PLND, the independent variables in the preoperative data set, including BMI, should be roughly the same in both the control and the case (PIF) groups. However, this study claims that BMI is a risk factor.  This conclusion conflicts with the goals of a randomized trial. Without relevant explanations, the conclusion is questionable.

2. Since the ProLy is a multi-center RCT, the data analyses should take the potential "center effect" into account. However, in the data analysis, the center effect is totally ignored. Needs explanation or re-do data analysis.

Minor:

1. In Table 1~3, the independent variables collected from pre-operative, peri-operative, and post-operative should be grouped together for better understanding.

2. The variable names in Table 2 are inconsistent with those in Tables 1 and 3.

3. The statistical methods mentioned in lines 98--101 are not applied in data analysis.

4. There are several unexpected symbols in the sentences. For example, an extra "%" in line 22, a "_" in line 87, and a ":", instead of an ".",  in line 97. Some other spelling check is needed.

Author Response

Response to Reviewer 1 Comments

Major revisions

Point 1: In the trial, since the randomization is conducted after the completion of RARP and PLND, the independent variables in the preoperative data set, including BMI, should be roughly the same in both the control and the case (PIF) groups. However, this study claims that BMI is a risk factor.  This conclusion conflicts with the goals of a randomized trial. Without relevant explanations, the conclusion is questionable.

Response 1: We thank the reviewer for this comment. The randomization process of the ProLy study is described in detail in the open source article “Bilateral Peritoneal Flaps Reduce Incidence and Complications of Lymphoceles after Robotic Radical Prostatectomy with Pelvic Lymph Node Dissection – Results of the Prospective Randomised Multicentre Trial ProLy”, DOI: 10.1097/JU.0000000000002693. Randomization was performed after completion of RARP and PLND and no statistical significant difference was found between groups (see Table 1 in the abovementioned article: median body mass index: group A (PIF) 26 kg/m2 (25-29) vs. group B (no PIF) 27 kg/m2 (25-30). The (non published) p-value for BMI was p=0.62).

In this present article, we compared patients with lymphocele vs. patients without lymphocele and found BMI to be an independent risk factor for lymphocele formation.

Point 2: Since the ProLy is a multi-center RCT, the data analyses should take the potential "center effect" into account. However, in the data analysis, the center effect is totally ignored. Needs explanation or re-do data analysis.

Response 2: Thank you for this comment. To rule out possible center effects, we performed a center-dependent analysis with respect to lymphoceles and symptomatic lymphoceles. We found no significant differences between the participating study centers. We added this information in our manuscript (see page 2, line 113 ff.).

For further information, please see the following added Tables.

Univariate analysis for the occurrence of lymphocele.

Univariate

95% KI

Variable

OR

Lower Bound

Upper Bound

p

Center 1 vs. Center 4

0.892

0.497

1.601

0.7025

Center 2 vs. Center 4

0.783

0.448

1.369

0.3906

Center 3 vs. Center 4

0.744

0.295

1.874

0.5298

All confidence intervals enclosed 1. The overall center effect had a p-value of 0.822. Thus, no center effect could be detected. Also, when another center was chosen as reference, all relevant p-values were > 0.05. For this reason, the center effect was not examined in the multivariate model.

Univariate analysis for the occurrence of symptomatic lymphoceles.

Univariate

95% KI

Variable

OR

Lower Bound

Upper Bound

p

Center 1 vs. Center 4

1.117

0.326

3.825

0.8599

Center 2 vs. Center 4

1.148

0.355

3.712

0.8174

Center 3 vs. Center 4

2.893

0.678

12.345

0.1513

All confidence intervals enclosed 1. The overall center effect had a p-value of 0.420. Thus, no center effect could be detected. Also, when another center was chosen as reference, all relevant p-values were > 0.05.

The center effect was not included in the multivariate model.

Minor:

Point 3: In Table 1~3, the independent variables collected from pre-operative, peri-operative, and post-operative should be grouped together for better understanding.

Response 3: The independent variables were grouped according to your suggestion. Please see the updated versions of Tables 1-4. We highlighted the changes in red.

Point 4: The variable names in Table 2 are inconsistent with those in Tables 1 and 3.

Response 4: Tables 1 and 3 show frequency distributions of potential risk factors for (significant) lymphoceles, depending on whether a lymphocele had occurred or not. Tables 2 and 4 show the results of the multivariate analysis. Therefore, the names of the variables differ, as the more detailed description explains the type of conducted analysis.

Point 5: The statistical methods mentioned in lines 98--101 are not applied in data analysis.

Response 5: We apologize and added these informations. You can now determine which test was used for the individual variable. Please see the updated versions of Tables 1 and 3.

Point 6: There are several unexpected symbols in the sentences. For example, an extra "%" in line 22, a "_" in line 87, and a ":", instead of an ".",  in line 97. Some other spelling check is needed.

Response 6: We have checked the text again for spelling errors and removed the described, accidentally added symbols

Reviewer 2 Report

Nice well written paper with correct statistical analysis.

However, there are several issues arising from the paper:

a) use of low molecular weight heparin in the investigated population of patients;

b) ASA system classification: stratify ASA 3 from ASA 2 and 1;

c) the median number of counted lymph nodes is low and this may impact results and conclusions of the study;

d) clinical staging, ISUP grade groups and clinical risk classes classification are missing;

e) the impact of surgeon who performed the procedures;

f) Clavien-Dindo complications have not been reported as treatment of symptomatic lymphoceles;

f) these issues need to be discussed as why obese patients were more likely to have symptomatic lymphoceles, as well.

I suggest to refer to the following papers while discussing the results:

  • DOI: 10.1007/s11701-018-0824-3

  • DOI: 10.1007/s11701-022-01505-7

  •  

Author Response

Response to Reviewer 2 Comments

Nice well written paper with correct statistical analysis.

However, there are several issues arising from the paper:

Point 1: use of low molecular weight heparin in the investigated population of patients

Response 1: Thank you for this comment. Application of low molecular weight (LMW) heparin was standardized by the study protocol and had to be given for 4 weeks postoperatively and had to be applied subcutaneously in the abdominal fat tissue. Due to only minimal deviations from the study protocol, no meaningful regression analysis could be performed. We added this information in the manuscript (see lines 99 f. and lines 117 f.).

Point 2: ASA system classification: stratify ASA 3 from ASA 2 and 1

Response 2: We added this analysis in our tables 1-4. 42 patients without lymphocele had an ASA score = 3 (12.7%). 18 patients with lymphocele had an ASA score = 3 (14.0%). The p-value of the chi-square test provided a p-value of 0.718. In addition, we could not identify the ASA score as a risk factor for lymphoceles in the multivariable analysis.

With regard to symptomatic lymphoceles, 59 patients had an ASA score = 3 (12.6%). 2 patients with symptomatic lymphocele had an ASA score = 3 (7.7%, p=0.403). No independent predictor for this variable could be found in multivariate analysis, either.

Point 3: the median number of counted lymph nodes is low and this may impact results and conclusions of the study

Response 3: The median number of counted lymph nodes in the ProLy study was 14 for both groups1 (IQR: group A 11-18, group B 11-19, respectively). The pelvic lymph node dissection was performed as extended pelvic lymph node dissection. Other randomized studies analyzing extended pelvic lymph node dissection found comparable median lymph node yields between 14 and 17.2-4 We added this information in the “Limitations” section of our manuscript.

Bibliography:

1    Gloger et al. “Bilateral Peritoneal Flaps Reduce Incidence and Complications of Lymphoceles after Robotic Radical Prostatectomy with Pelvic Lymph Node Dissection – Results of the Prospective Randomised Multicentre Trial ProLy”, DOI: 10.1097/JU.0000000000002693.

2 Bründl et al. “Peritoneal Flap in Robot-Assisted Radical Prostatectomy”, DOI: 10.3238/arztebl.2020.0243.

3   Touijer et al. “Limited versus Extended Pelvic Lymph Node Dissection for Prostate Cancer: A Randomized Clinical Trial”, DOI: 10.1016/j.euo.2021.03.006.

  1. Lestingi et al. “Extended Versus Limited Pelvic Lymph Node Dissection During Radical Prostatectomy for Intermediate- and High-risk Prostate Cancer: Early Oncological Outcomes from a Randomized Phase 3 Trial, DOI: 10.1016/j.eururo.2020.11.040.”

Point 4: clinical staging, ISUP grade groups and clinical risk classes classification are missing

Response 4: Clinical staging and grading information are reported by pT, pN stages and ISUG grading groups (see Tables 1-4). No risk factors for lymphocele or symptomatic lymphocele could be identified for these variables.

Point 5: the impact of surgeon who performed the procedures

Response 5: A total of 19 surgeons (Bochum 3, Gronau 11, Homburg 1 and Leipzig 4) performed the procedures of the ProLy study. Unfortunately, the individual surgeon who performed the procedures was not captured in our database. Instead, we looked for potential center effects which could be excluded prior to further analysis. We added this information in our manuscript ( see page 2, line 113 ff.).

Point 6: Clavien-Dindo complications have not been reported as treatment of symptomatic lymphoceles

Response 6: Thank you for this comment. The Clavien-Dindo classification for the ProLy study has been published elsewhere and was stratified in total, lymphocele-dependent or lymphocele-independent complications.1 Symptomatic lymphoceles were mostly treated by percutaneous drainage placement and/or laparoscopic fenestration and caused Clavien 3a/3b complications. Lymphoceles were one of the main reasons for high grade Clavien Dindo complications. The aim of this present study was to find potential risk factors for lymphoceles and symptomatic lymphoceles, respectively, but not for grade 3 Clavien Dindo complications.

1    Gloger et al. “Bilateral Peritoneal Flaps Reduce Incidence and Complications of Lymphoceles after Robotic Radical Prostatectomy with Pelvic Lymph Node Dissection – Results of the Prospective Randomised Multicentre Trial ProLy”, DOI: 10.1097/JU.0000000000002693.

Point 7: these issues need to be discussed as why obese patients were more likely to have symptomatic lymphoceles, as well.

Response 7: Thank you for your comment. We edited the manuscript accordingly. Please see the edited text in the manuscript (see page 6, lines 207 ff.).

These patients often suffer from impaired wound healing and surgeons face more complex conditions in obese patients, as longer surgery duration is required to obtain equivalent surgical results

Point 8: I suggest to refer to the following papers while discussing the results:

  • DOI: 10.1007/s11701-018-0824-3
  • DOI: 10.1007/s11701-022-01505-7

Response 8: Thank you for your suggestion. We have discussed the results in light of these two articles and added citations accordingly (see page 6, lines 209 ff. and see page 7, lines 244 ff.).

Round 2

Reviewer 1 Report

I thank the authors' responses and has no further question.

Reviewer 2 Report

The authors have appropriately addressed the questions raised and the manuscript can be accepted in its current form.